# A reproducible approach for the use of aptamer libraries for the identification of Aptamarkers for brain amyloid deposition based on plasma analysis

**Cathal Meehan, Soizic Lecocq, Gregory Penner** *

NeoVentures Biotechnology Europe SAS, Villejuif Bio Park, Villejuif, France

* gpenner@neoneuro-aptamers.com

**Data Availability Statement:** All data utilized in this study were obtained from the Australian Imaging, Biomarkers and Lifestyle (AIBL) study and have been anonymized at the source. The AIBL study

## Abstract

An approach for the agnostic identification and validation of aptamers for the prediction of a medical state from plasma analysis is presented in application to a key risk factor for Alzheimer's disease. brain amyloid deposition. This method involved the use of a newly designed aptamer library with sixteen random nucleotides interspersed with fixed sequences called a Neomer library. The Neomer library approach enables the direct application of the same starting library on multiple plasma samples, without the requirement for pre-enrichment associated with the traditional approach. Eight aptamers were identified as a result of the selection process and screened across 390 plasma samples by qPCR assay. Results were analysed using multiple machine learning algorithms from the Scikit-learn package along with clinical variables including cognitive status, age and sex to create predictive models. An Extra Trees Classifier model provided the highest predictive power. The Neomer approach resulted in a sensitivity of 0.88. specificity of 0.76. and AUC of 0.79. The only clinical variables that were included in the model were age and sex. We conclude that the Neomer approach represents a clear improvement for the agnostic identification of aptamers (Aptamarkers) that bind to unknown biomarkers of a medical state.

## Introduction

The diagnosis of complex diseases, including dementia, psychiatric disorders, auto-immune disorders, and gastro-intestinal diseases, remains a significant challenge when relying on blood-based approaches. Various "omics" approaches, such as genomics, transcriptomics, proteomics and metabolomics have encountered limitations in addressing this issue. Firstly, genomics and transcriptomics-based strategies struggle to provide accurate predictions for complex diseases due to the intricate nature of genetic risk factors. The complexity arises from the involvement of numerous genetic variants, each with a modest individual effect, having positive or negative correlations with the condition, making it challenging to pinpoint causative factors or establish predictive models. Additionally, transcriptomics fails to correlate gene

protocols were approved by the institutional ethics committees of Austin Health, St Vincent's Health, Hollywood Private Hospital and Edith Cowan University. Written informed consent was obtained from all participants. The data that support the findings of this study are available from the AIBL study and are supplied in the Supplementary information.

**Funding:** The study was funded by NeoVentures Biotechnology Europe SAS and the Alzheimer's Drug Discovery Foundation (project entitled "Blood diagnostic test for AD: aptamer deep biomarker fingerprinting", reference # GDAPB-201808-2016228) through a grant in 2020. During the course of this study, NeoVentures Biotechnology Europe was financially supported by NeoVentures Biotechnology Inc. The funders had a role in the study design, data analysis, decision to publish, and preparation of this manuscript. S. Lecocq and C. Meehan are employees of NeoVentures Biotechnology Europe. G. Penner is an employee of NeoVentures Biotechnology Inc.

**Competing interests:** The authors have read the journal's policy and have the following competing interests: This study was funded by Neoventures Biotechnology Europe SAS and the Alzheimer's Drug Discovery Foundation (project entitled "Blood diagnostic test for AD: aptamer deep biomarker fingerprinting" - GDAPB-201808-2016228). CMeehan and SLecocq are paid employees of Neoventures Biotechnology Europe SAS. GPenner is a paid employee of NeoVentures Biotechnology Inc. GPenner and another partner not mentioned in this study, Ximena Vedoya, privately own all shares in Neoventures Biotechnology Inc. and Neoventures Biotechnology Europe SAS. Patent applications have been submitted globally describing both the Neomer application and the Aptamarker approach. This does not alter our adherence to PLOS ONE policies on sharing data and materials.

expression from these individual genetic factors to disease onset or progression due to technical variation introduced during sample preparation, library construction, and sequencing [1]. Both of these approaches are limited by the gap between genetic variation, transcript abundance and protein abundance as a function of translation. Neither of these approaches provide information about molecular complexes or post-translational modifications.

Further downstream, proteomics and metabolomics, pose another set of challenges. These platforms confront substantial variability, both within individuals due to dynamic fluctuations in protein levels or metabolite concentrations over time, and across different subjects (inter-individual) due to genetic diversity, lifestyle, or environmental factors [2]. Proteomics and metabolomics are also both limited in their ability to detect complexes and to quantify the relative abundance of molecules. Uncharacterized forms of both metabolites and proteins, as well as protein complexes, further limit the ability of these approaches to form predictive models based on current scientific literature. Proteins and metabolites in life often exist in complexes, many proteins exist and function as members of non-covalently bound homo or hetero-mers of varying composition, while metabolites due to their inherent hydrophobicity often exist in the blood stream as non-covalent complexes with protein. The existence of complexes is not directly measured by either proteomics or metabolomics, it is at best inferred.

There is a need for an agnostic approach to biomarker identification that is not constrained by these limitations. The concept of using aptamer libraries for the agnostic discovery of biomarkers for predicting disease states has been attempted by ourselves [3, 4] and others but was also limited by the following constraints.

1. The need to immobilize target material in order to parse aptamers that bind from aptamers that do not bind.

2. The need for reiterative selection cycles in order to enrich aptamer sequences from single unique copies in the naïve library. The reiterative selection process implicitly introduces PCR bias into the selection process and is a cause for variation between samples.

3. The inability to apply the same set of sequences in a naïve library to different biofluid or tissue samples.

In this paper we will describe how we have combined innovative approaches that overcome these constraints with an aptamer library application for the diagnosis of brain amyloid deposition (a risk factor for Alzheimer's disease) in blood.

We have overcome the need to immobilize targets in order to separate aptamers in a library that bind from those that do not with an approach called FRELEX [5]. This approach relies on immobilized antisense oligonucleotides on gold nanoparticles to compete with targets in blood for the binding of all aptamers in a library. A proportion of each aptamer sequence that does not bind to a target will hybridize to the immobilized antisense and can be removed with centrifugation. That portion of any aptamer sequence that binds to a target will remain in the supernatant and is recovered and submitted for next generation sequencing (NGS) analysis. The more any given aptamer is bound to targets in selection the higher it's relative frequency will be in NGS analysis.

We overcame the latter two constraints listed above, the need for reiterative selection and the inability to apply the same aptamer sequences to different biological samples by reinventing aptamer selection. A key constraint with the traditional SELEX method is the contiguous nature of the random region [6]. This implicitly restricts structural diversity resulting in a need to design libraries with large numbers of random nucleotides. The standard SELEX library design involves 40 contiguous nucleotides (nt) in the random region. This represents a total of 1.2E24 possible sequences. In practical terms the upper limit in regard to the number

of sequences that can practically be applied to a sample is 1E15 to 1E16. This means that it is not practical to apply aptamer libraries in selection that contain multiple copies of each possible sequence. It is also not possible to apply the same library of sequences to different targets. To overcome this constraint we designed an aptamer library with 16 random nt interspersed with fixed sequences, as described in Fig 1. The key to this design is that there is minimal capacity for hybridization among the fixed sequences. With 16 random nt the total number of possible sequences is 4.29E9. This approach enables the use of 4E12 sequences such that there is an average of 1,000 copies of each possible sequence in the naïve library, and the same naïve library sequences can be applied to different biological samples. We refer to this novel design of an aptamer library as a neomer library.

The next issue to be solved was how to characterize the frequency of all 4.29E9 sequences cost-effectively in NGS analysis. Our solution was to include a restriction site in the middle of the library sequence. The presence of this restriction site drives the only occurrence within the fixed sequences of self-hybridization. After selection, the library is amplified and then this restriction site is used to cut the library into two modules. Each module contains eight random nucleotides representing a total of 65,536 possible sequences. Each module is prepared separately for NGS analysis and in a single NGS run it is possible to determine the frequency of each of all possible sequences.

The frequency of each of the original 4.29E9 possible sequences is obtained for each selection by multiplying the frequencies of each of the possible 65,536 sequences in one module by

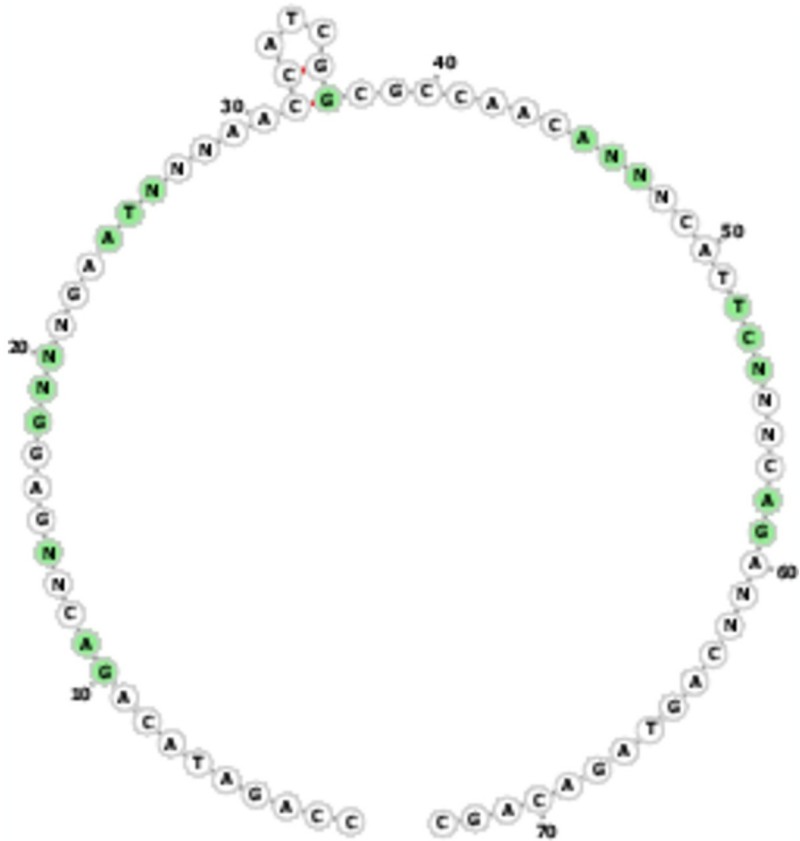

**Fig 1. Predicted secondary structure of fixed sequences in Neomer library.**

the frequencies in the other module through the use of an outer product matrix. This mathematical approach is synonymous to the concept of a Punnett square used in genetics.

We refer to the aptamers within these libraries used for agnostic selection as Aptamarkers. We applied this neomer approach to identify Aptamarkers for brain amyloid deposition in plasma with 390 individuals from the Australian Imaging, Biomarker and Lifestyle (AIBL) cohort [7, 8].

Alzheimer's disease is a neurodegenerative process that is thought to be caused by a three-step process [9]. The first step is the accumulation of amyloid plaques in the brain, followed by Tau tangles within neurons and finally neuronal death. The presence of high brain amyloid is considered by clinicians in combination with memory tests in their definition of the disease in patients presenting memory symptoms. The use of positron emission tomography (PET) scans or the evaluation of Abeta 42 to Abeta 40 peptide ratios in cerebrospinal fluid [10] are the current gold standards for brain amyloid diagnosis. Unfortunately, both diagnostic procedures are invasive and expensive. The need for a simple blood test as a first line screen has been suggested as desirable by several sources [11, 12].

Recent computational tools and algorithms have been developed for predicting various properties or functions related to proteins, specifically focusing on antimicrobial peptides (AMPs) and antiviral peptides (AVPs). These predictors, such as DeepAVP-TPPred [13], Deepstacked-AVPs [14], iAFPs-Mv-BiTCN [15], pAtbp-EnC [16], and pAVP_PSSMDW-T-EnC [17], demonstrate the potential of computational methods for identifying functional peptides and proteins. While these approaches could potentially be adapted for the prediction of peptides or proteins associated with Alzheimer's disease and brain amyloid deposition, they have certain limitations. First, these methods rely on known biomarkers or peptide sequences for training the models, limiting the ability of these platforms and techniques to identify novel or unknown biomarkers. Second, the complexity of Alzheimer's disease pathology and the multifactorial nature of its progression may not be fully captured by focusing solely on peptide sequences. Third, the translation of these computational models into clinical practice would require extensive validation and the integration of multi-modal data, such as clinical and imaging information, to improve their diagnostic accuracy and utility.

In contrast, the Neomer library design presents a novel, agnostic approach for identifying blood-based Aptamarkers predictive of brain amyloid deposition. This approach leverages the differential frequency of aptamers between samples with and without the disease condition, enabling the discovery of Aptamarkers without relying on prior knowledge of specific biomarkers. By applying the Neomer library to plasma samples from individuals with high and low brain amyloid deposition, we aimed to identify Aptamarkers that could serve as a cheap, non-invasive, and predictive blood-based test for Alzheimer's disease.

We applied a neomer library to plasma samples from twenty individuals (ten high brain amyloid, ten low brain amyloid) balanced for other clinical variables and characterized the frequencies of all 4.29E9 sequences across all samples by NGS analysis. We identified four Aptamarkers that were preferentially enriched on blood samples from individuals with high brain amyloid deposition (HAM) and four sequences that were preferentially enriched on blood samples from four individuals with low brain amyloid (LAM). We designed specific primer sequences for each Aptamarker such that we could amplify them individually from a pool. The pool of eight Aptamarkers were applied to 10 uL of plasma from 390 samples (AIBL cohort) in the presence of antisense oligonucleotides (Fig 2) with the capacity to hybridize to them. The antisense oligonucleotides were immobilized on gold nanoparticles and that portion of the Aptamarker pool that hybridized to them was removed with a centrifugation step. The remaining supernatant was used as a template for qPCR analysis.

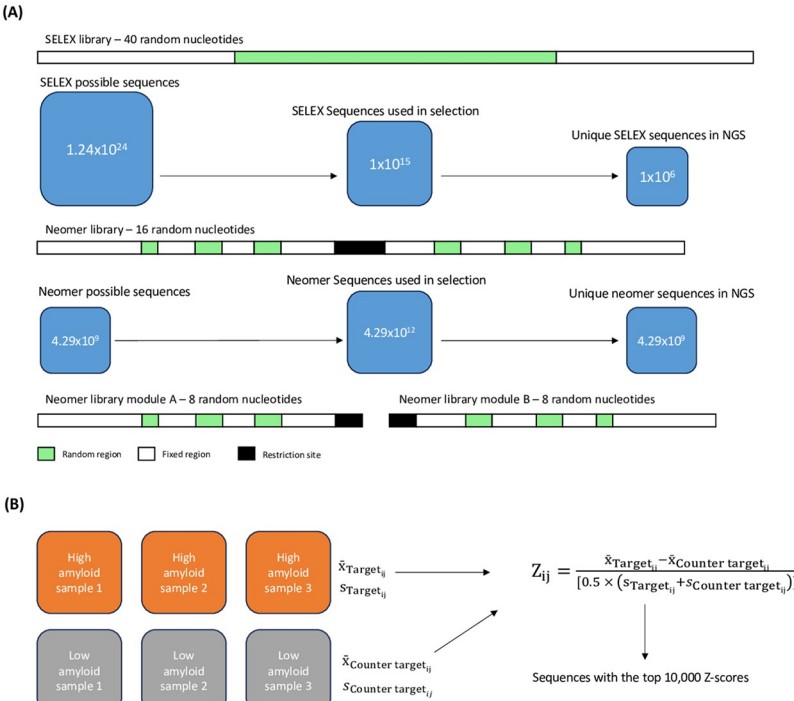

**Fig 2. Schematic representation of the application of neomer aptamers to plasma samples.**

Prior to processing the samples we predefined 291 of the samples as training samples and 99 as test samples. Machine learning (ML) analysis of the qPCR results and clinical variables was carried out using a variety of different algorithms on the training samples. ML models that were assessed for sensitivity, specificity and area under the receiving operator curve (AUROC) and were tested for predictive capacity on the test samples.

ML analysis of the Neomer libraries with eight Aptamarkers and age resulted in an Extra Trees Classifier model with a sensitivity of 0.88. specificity of 0.76. overall accuracy of 0.82 and an area under the AUROC curve of 0.79. The predictive power of the Neomer/Aptamarkers did not include cognitive status as a clinical variable, indicating that this was not a significant criteria for this test. A key advantage of the method as described is the agnostic nature of the analysis. This same approach can be applied to develop Aptamarkers for any disease. The method can also be applied to detect differences in blood related to response to treatment. or predisposition to side effects.

## Materials and methods

### Selection of plasma for Neomer selection

Plasma from 10 cognitively normal (CN) individuals for both high and low brain amyloid deposition from the AIBL cohort were selected to maximise the separation of the two groups while balancing for other clinical variables (S1 Table). Cognitively normal plasma samples were chosen to ensure only biomarkers implicated in brain amyloid deposition were being selected and not pathological factors involved in memory impairment.

### Ethics statement

The AIBL study protocols were approved by the institutional ethics committees of Austin Health, St Vincent's Health, Hollywood Private Hospital and Edith Cowan University. Written informed consent was obtained from all participants.

### Neomer selection and predictive modelling

The ssDNA Neomer library used for selection was composed of a 16 random interspersed by fixed sequences.

5'CCAGATACAGACNNGAGGNNNNGAATNNNAACCATCGGCGCCAACANNNCATTCNNNC AGANNCAGTAGACAGC 3' (IDT, Belgium).

This template was selected to limit the contiguous nature of traditional aptamer libraries and to maximise the structural diversity of the aptamers. A prediction of the secondary structure formed by this library is provided in Fig 1 based on the use of the RNAfold program [18] as visualized with Forna structure software [19].

We also conjugated single stranded (ss) DNA antisense sequences (/5ThioMC6-D/ AAAACAAAGTCTACTTGTTGGTTCTGTAT) with homology to fixed sequences in the neomer library on 40 nm Gold Nanoparticles (GNPs—Cytodiagnostics). This antisense was designed to have complementary binding to fixed sequences in the Neomer library. The thiolated antisense was reduced with TCEP following IDT specifications. Reduced antisense was then conjugated with GNPs following supplier specifications and dissolved in 200 μL 0.1X of phosphate buffer saline (PBS) (8.0 mM Na2PO4, 1.4 mM KH2PO4, 136 mM NaCl, 2.7 mM KCl, pH 7.4). This concentration of the antisense in solution on the GNPs was measured with a NanoDrop instrument in order to ensure that an antisense concentration of 0.24 μM was used for each sample.

In the first step of FRELEX employed in this study, 2.57E13 sequences from the neomer aptamer library were snap cooled by heating the library to 95 °C for 10 min followed by immediate immersion in ice bath. These libraries were then incubated with individual plasma sample in 50 μL of Selection Buffer (10 mM Tris, 120 mM NaCl, 5 mM MgCl2, 5 mM KCl) for 15 min at RT. This solution was then applied to the functionalized GNPs for 15 min at RT. The solution was centrifuged for 20 min at 3500 g. The supernatant was removed and purified using the Oligonucleotide Clean-up protocol of the Monarch PCR & DNA Clean-up kit, as described by NEB and eluted with 400 μL of de-ionised water.

After this selection, PCR amplification was used to double strand the library (dsDNA) for an appropriate number of cycles to create a clear band of approximately 5 ng of amplified DNA. All PCR procedures were carried out according to standard molecular biology protocols and under the following conditions: 95°C for 5 min, 4 cycles at 95°C for 10 s, 35°C for 15s, 72°C for 30 sec, and then remaining cycles at 95°C for 10 s, 64°C for 15s and 72°C for 30 sec, followed by a final extension at 72°C for 5 min. Table 2 provides the primer sequences used for all steps described here. The underlined sequence in Table 1 is a hex code that was used for library parsing from the fasta file. The sequence shown is only one example, we used 21 different hex codes for this analysis. We used the same hex code for module A and module B in all cases.

It is important to maintain the average expected copy of each sequence at a level above 100 copies throughout the NGS preparation procedure. To ensure this we performed the Oligonucleotide Clean-up protocol of the Monarch PCR & DNA Clean-up kit (NEB) on 10 PCR reaction tubes. The 40 μL volume of elution was used as a template for restriction with KasI enzyme (NEB) at 37°C for 60 min followed by 20 min at 65°C to obtain two distinct modules

**Table 1. Sequences of primers used to process Neomer libraries for NGS analysis.**

| Name of primer | Primer sequence |
| --- | --- |
| Full Library Fwd | 5' CAA ATA CGT ATG AGG TCG CTC GTT CCC AGA TAC AGA C 3' |
| Full library Rvs | 5' TAA TAC GAC TCA CTA TAG GGA TAA TGC TGT CTA CTG 3' |
| NGS-1A Fwd | 5' CCC TAC ACG ACG CTC TTC CGA TCT ATC ACG CAA ATA CGT ATG AGG TCG CTC GTT C 3' |
| NGS-1A Rvs | 5' GGT CAG ACG TGT GCT CTT CCG ATC GGG GCG CCG ATG GTT 3' |
| NGS-1B Fwd | 5' CCC TAC ACG ACG CTC TTC CGA TCT ATC ACG GCG CCA ACA 3' |
| NGS-1B Rvs | 5' GGT CAG ACG TGT GCT CTT CCG ATC GGG TAA TAC GAC TCA CTA TAG GGA TAA TGC TGT CTA CTG 3' |

(A and B). The two modules were treated in the same way, but separately for Next Generation Sequencing (NGS) preparation. The restricted products were diluted 10X prior to next step.

The amplified product from NGS preparation was purified using the Monarch PCR & DNA Clean-Up kit prior to be sequenced. In addition, we processed a buffer sample containing no blood in each analysis, as a further test.

All libraries were sequenced using Illumina NovaSeq at the TCAG facility (Hospital for Sick Children, Toronto, Canada). Fastq files obtained upon sequencing were converted into a Fasta format, and the copy number of each of the possible 65,536 sequences in each of Module A and Module B were characterized using a proprietary Python script that we have developed for this application. The frequency of each Module sequence was determined by dividing copy number by the total number of reads for that Module. The frequency of each of the 4.29 billion sequences in the original selected library was then estimated through an outer product calculation of the Module A and Module B frequencies.

The average frequency for each of these 4.29 billion sequences was determined for the ten high amyloid samples and the ten low amyloid samples. The standard deviation of each of these averages was also determined. The average of the low amyloid samples for each sequence was subtracted from the average for the same sequence across the high amyloid samples. This value was divided by the average of the standard deviation from both sets of samples to generate a Z score. The top 10,000 sequences in terms of Z score were identified.

We then identified the top four sequences within these 10,000 based on the separation of all the ten high samples from all the ten low samples.

We also ran this analysis by subtracting the average of each sequence with the high amyloid samples from the average of the same sequence averaged across the low amyloid samples. The top 10,000 sequences based on Z scores was also identified as low amyloid sequences and the top four identified based on separation of all ten low samples from all ten high samples.

The sequences of these eight Aptamarkers is provided in Table 2. Specific reverse primers were designed for each of these sequences and tested to ensure that in conjunction with the same forward primer (GTT CCC AGA TAC AGA C) in each case they specifically amplified the desired sequence.

## Screening of Aptamarkers

Each sample was analyzed in two well qPCR replicates. Following qPCR analysis, the data was reviewed according to quality control standards with a coefficient of variance of 6 between replicates, only data passing this standard was retained. The raw fluorescence data was analyzed to identify the point at which fluorescence increased to six times the base level. The base level was determined in the first five cycles. The point between the PCR cycles was determined by linear regression analysis. This provided a Cq value. An efficiency value (E) for each sample was determined based on identification of the linear portion of the amplification curve above

**Table 2. Sequences of Aptamarkers identified by the Neomer process and specific reverse primer sequences for their amplification.**

| Aptamer name | Aptamer sequence | Reverse primer sequence |
|---|---|---|
| HAM_2753 | CCA GAT ACA GAC TCG AGG ACT GAA TCG GAA CCA TCG GCG CCA ACA AGA CAT TCA TAC AGA AAC AGT AGA CAG C | TTT CTG TAT GAA TGT CT |
| HAM_6700 | CCA GAT ACA GAC TCG AGG TAA GAA TGG GAA CCA TCG GCG CCA ACA CAG CAT TCA ATC AGA GAC AGT AGA CAG C | TCT CTG ATT GAA TGC TG |
| HAM_6968 | CCA GAT ACA GAC CAG AGG CGT GAA TTT CAA CCA TCG GCG CCA ACA CCC CAT TCC TGC AGA TAC AGT AGA CAG C | TAT CTG CAG GAA TGG GG |
| HAM_8505 | CCA GAT ACA GAC AGG AGG GCG GAA TTC CAA CCA TCG GCG CCA ACA TTT CAT TCT AAC AGA CAC AGT AGA CAG C | TGT CTG TTA GAA TGA AA |
| C-LAM_1 | CCA GAT ACA GAC TAG AGG TAT GAA TAG AAA CCA TCG GCG CCA ACA ATT CAT TCA TTC AGA TGC AGT AGA CAG C | CAT CTG AAT GAA TGA AT |
| C-LAM_168 | CCA GAT ACA GAC AAG AGG ACC GAA TGT CAA CCA TCG GCG CCA ACA TTA CAT TCA TTC AGA TTC AGT AGA CAG C | AAT CTG AAT GAA TGT AA |
| C-LAM_262 | CCA GAT ACA GAC CAG AGG TTT GAA TTC CAA CCA TCG GCG CCA ACA TTA CAT TCC TCC AGA ATC AGT AGA CAG C | ATT CTG GAG GAA TGT AA |
| C-LAM_2709 | CCA GAT ACA GAC TTG AGG TTC GAA TCT CAA CCA TCG GCG CCA ACA TGA CAT TCC TTC AGA TTC AGT AGA CAG C | AAT CTG AAG GAA TGT CA |

the Cq value. The average of all Cq and E values within a given qPCR experiment were calculated. This average was divided by a fixed value to define a normalization factor for each qPCR run. Both the Cq and E values were divided by this normalization factor by qPCR set. The Cq and E values were averaged across replicates and the relative amount of template (RT) was then determined with the following formula—$(1 \ E)^{Cq}$.

Each individual RT value was divided by the sum of all eight RT values per sample to define a proportion of the RT value within each sample (RTp). This value was used for model building.

## Dataset preprocessing

The dataset model comprised 390 samples, with variables for all 8 aptamers and the same 3 clinical variables. Both were preprocessed for zero-mean scaling, to make them suitable for ML analyses.

## Feature selection using default parameters

For each classification algorithm, a model with default parameters was applied to calculate the importance of individual features. The algorithms from the scitkit-learn package [20] and others were used, these include Logistic Regression, Decision Trees, Random Forests, Gradient Boosting, k-Nearest Neighbors, Support Vector Machines, Gaussian Naive Bayes, Multi-Layer Perceptron, XGBoost [21], Ridge Classifier, and ExtraTrees. Based on the importance scores obtained, features were ranked. In a sequential approach, the least significant features were excluded, thus generating reduced training datasets.

## Hyperparameter tuning

For each iteration of feature selection, hyperparameter tuning was conducted. A predefined dictionary of hyperparameter grids was set for each algorithm under evaluation. The GridSearchCV technique, complemented by a 4-fold cross-validation, was used to pinpoint the optimal hyperparameters for each model, with accuracy serving as the optimization criterion.

## Model evaluation

Models were subsequently trained on the reduced datasets obtained post-feature selection. During each model iteration, evaluative metrics such as accuracy, sensitivity, specificity, and AUROC were gauged. A 0.5 threshold was used to determine classification for either high or low brain amyloid deposition.

## Results

The ssDNA Neomer library used in our selection was a 16-base random sequence interspersed with fixed sequences. This library was designed to optimise secondary structure diversity through the placement of regions of random sequence and by the lack of capacity of the fixed sequences to hybridize with each other (Fig 1). This means that structural diversity is entirely driven by the random nucleotides.

During NGS preparation, it was crucial to maintain an average expected sequence copy above 100. The amplified products for NGS were purified before sequencing. A buffer sample was incorporated in each analysis as a control. Following sequencing, a proprietary Python script was employed to determine the frequency of every sequence in Modules A and B. The frequency of each of the 4.29 billion sequences in the selected library was deduced through an outer product calculation.

From the analysis, we discerned the average frequency for all sequences across ten high and low amyloid samples. Z scores were then derived by considering the difference in averages and standard deviations in these sample sets (Fig 3). The top 10,000 sequences based on Z scores were identified and further refined to the top four sequences offering the best separation across sample sets when performing contrasts between high and low brain amyloid (Fig 4), and low and high brain amyloid deposition (HAM and LAM aptamers).

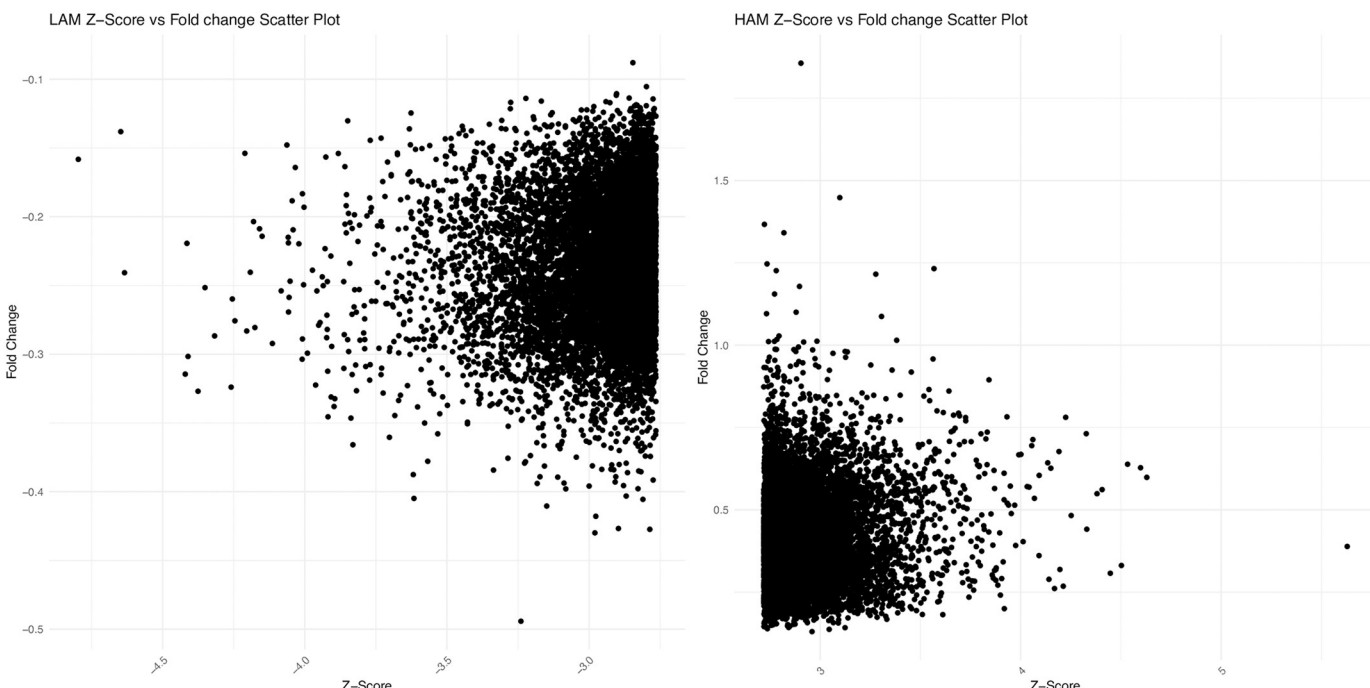

**Fig 3. Distribution of Z-score and fold values across contrasts.** A) Scatterplot of Z-score and fold values in the top 10000 aptamers in the high amyloid versus low amyloid contrast. B) Scatterplot of Z-score and fold values in the top 10000 aptamers in the low amyloid versus high amyloid contrast.

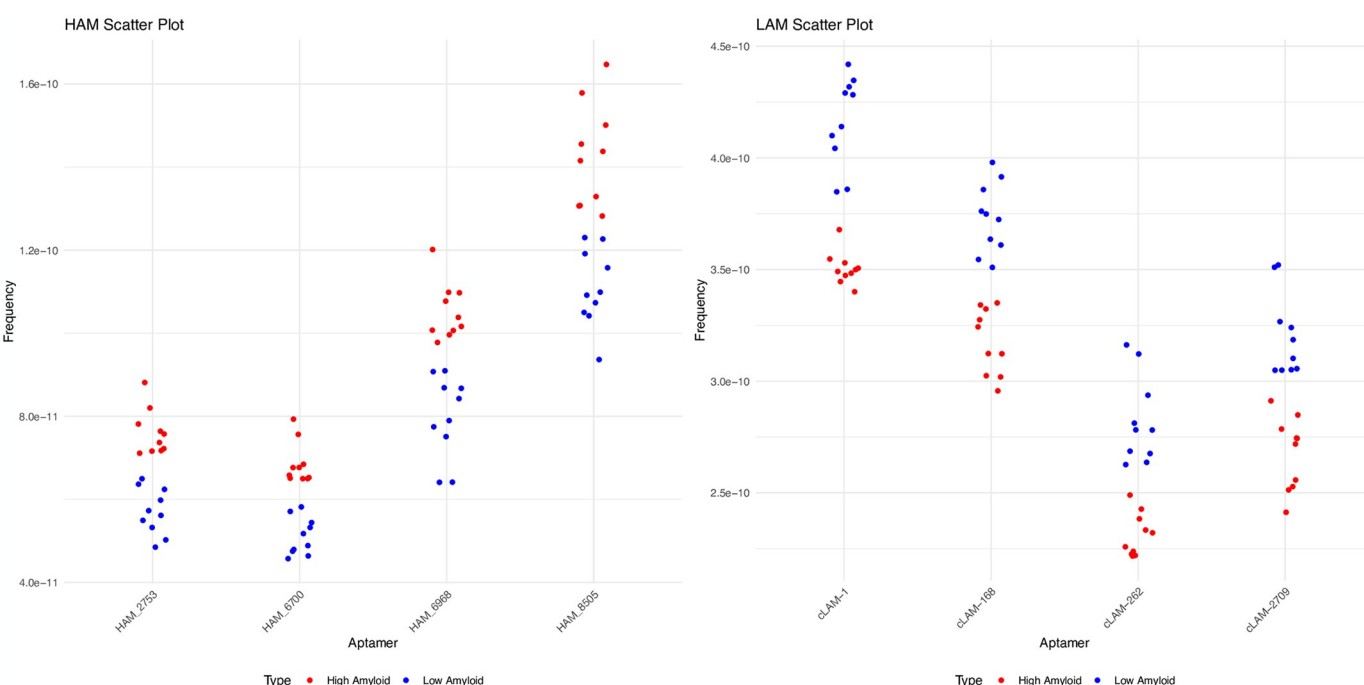

**Fig 4. Comparative analysis of samples utilizing the top 4 of high and low amyloid associated aptamers.** A) Scatterplot of the frequency of the top 4 selected high amyloid versus low amyloid contrast aptamers. B) Scatterplot of the frequency of the top 4 selected low amyloid versus high amyloid contrast aptamers. Each point corresponds to an individual sample.

It is interesting to note that the fold differences were higher with the high versus low amyloid contrast as compared to the low versus high contrast. The higher fold value for the high amyloid aptamers may indicate that these will be more predictive than the low amyloid ones.

In the qPCR analysis of Aptamarkers, we established a strict quality threshold with a coefficient of variance of 6 between replicates. Using the fluorescence data, we computed the Cq and efficiency values (E) for each sample, subsequently determining a normalization factor. Relative amounts of template (RT) were calculated, and from these, proportions of RT values per sample (RTp) were derived, forming the basis for model building.

We used all eight Aptamarkers as variables and the clinical variables sex, clinical status, and age. Clinical status was classified in the AIBL cohort as either mild cognitive impairment (MCI), subjective memory complaints (SMC) and no memory complaints (NMC). To objectively assess the clinical status, we grouped both SMC and NMC together as cognitively normal (CN).

We did not include ApoE genetic status in our primary model based on the understanding that this variable is not commonly applied in clinical practice in Europe. However, we did include this variable to assess whether our Aptamarkers were implicitly providing the predictive power of the ApoE allelic status. This was classified as either 0, 1, or 2 depending on the number of E4 alleles that were present in the allelic status of the patient. E2 alleles weren't considered in the classification here due to their low incidence in our samples from AIBL cohort, where they normally would be considered protective against brain amyloid deposition [22]. The distribution and variation of clinical variables in our samples can be found in S1 Fig.

A centiloid value of 30 was determined as the threshold for high and low brain amyloid deposition as per the recommendation of the AIBL clinicians and many other studies which have indicated this is a robust value to classify an established pathology [23].

**Table 3. Top models based on analysis of training and testing set samples using a suite of ML algorithms with sequential feature exclusion based on feature predictive importance.**

| Model | Excluded Features | Training Accuracy | Testing Sensitivity | Testing Specificity | Testing AUROC |
|---|---|---|---|---|---|
| ExtraTreesClassifier | Clinical Classification | 1 | 0.88 | 0.76 | 0.79 |
| ExtraTreesClassifier | Rvs c-LAM262 Clincal Classification, Sex | 1 | 0.76 | 0.76 | 0.78 |
| MLPClassifier | | 0.68 | 0.67 | 0.76 | 0.78 |
| LogisticRegression | Rvs c-LAM1, Rvs c-LAM168, Rvs HAM6700, Rvs HAM6968, Sex | 0.68 | 0.65 | 0.78 | 0.77 |
| ExtraTreesClassifier | | 1 | 0.76 | 0.7 | 0.77 |
| LogisticRegression | | 0.67 | 0.67 | 0.74 | 0.77 |
| ExtraTreesClassifier | Rvs c-LAM262, Rvs HAM6700, Clinical Classification, Sex | 0.99 | 0.78 | 0.76 | 0.77 |
| LogisticRegression | Rvs c-LAM168, Rvs HAM6700, Rvs HAM6968, Sex | 0.67 | 0.65 | 0.76 | 0.76 |

The dataset was preprocessed to zero-mean scaling, catering to ML applications. Feature selection employed default hyperparameters from each ML algorithm being trained and tested, sequentially excluding less significant features (features that provided less predictive capacity to the model). With each iteration, hyperparameter tuning was performed using the Grid-SearchCV technique and a 4-fold cross-validation. The models, trained on the reduced datasets, were evaluated based on multiple performance metrics. Table 3 provides a summary of the top algorithms that returned high predictive results on the test set.

Many of the top ML models that exhibited the highest fit on the observed data were tree-based ensemble methods. An ExtraTrees model without clinical classification that had 1.0 accuracy on the training set (ET-1) performed best, showing a test data sensitivity of 0.88, specificity of 0.76, overall accuracy of 0.82 and AUROC of 0.79.

The inclusion of ApoE status in the dataset resulted in a Random Forest Classifier model that included all clinical variables and Aptamarkers performing best, with a 1.0 accuracy on the training set (RF-1) with a test data sensitivity of 0.81, a specificity of 0.8, an overall accuracy of 0.82, and an AUROC of 0.81 (S3 Table). We interpret this result as meaning that the Aptamarkers are implicitly providing the predictive power of the ApoE allelic status. Full tables for both datasets used with the training data sensitivity and specificity and hyperparameters used can be found in S2 and S3 Tables.

Fig 5 displays the feature importances in the random forest model, which measure each feature's contribution to the predictions by averaging the decrease in impurity across all decision trees when the feature is used for splitting. Features with higher importance scores are more influential in the model's decision-making process.

It is intriguing that the Aptamarkers all exhibit relatively similar predictive capacity with each individual Aptamarker contributing about half the value of age as a variable. The sum of all the Aptamarker values represents more than 74% of the predictive capacity of the model. The receiver operator response analysis is provided in Fig 6.

The area under the curve of the receiver operator AUROC analysis provides a description of the expected robustness of the test. In this case, that value is 0.79.

## Conclusions

The application of the Neomer library approach addresses several limitations of the traditional SELEX method and results in the identification of Aptamarkers with high predictive capacity for brain amyloid deposition from plasma analysis. The agnostic nature of this approach means it may be applicable to developing blood-based diagnostic tests for many diseases and conditions.

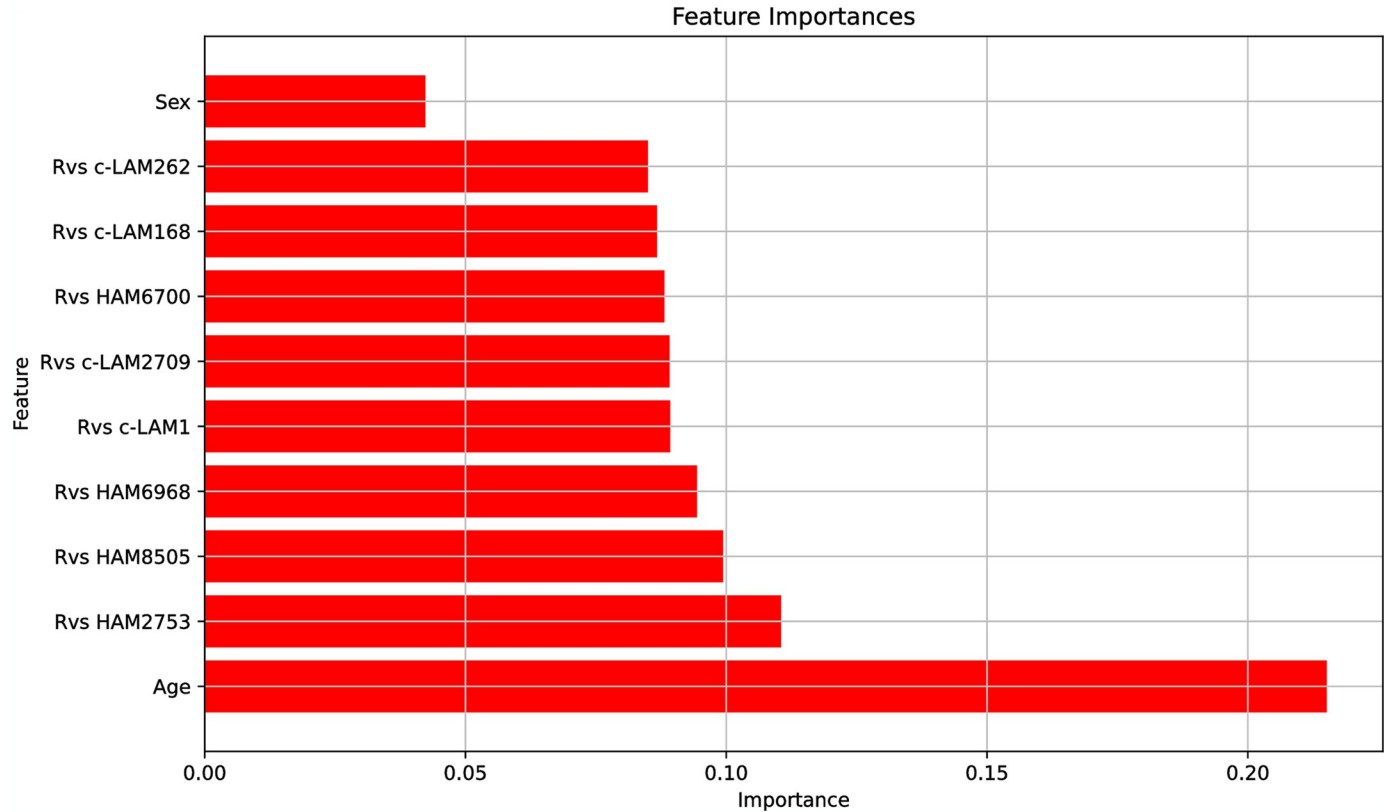

**Fig 5. Feature importance attributable to each variable in the ET-1 predictive model.**

The ET-1 model utilizing the 8 Aptamarkers identified through the Neomer process, along with age as a clinical variable, achieved a sensitivity of 0.88, specificity of 0.76, accuracy of 0.82, and AUROC of 0.79 on the test set. Notably, the Aptamarkers alone accounted for over 74% of the model's predictive power, highlighting their importance. The inclusion of ApoE status didn't improve model performance, suggesting the Aptamarkers may be implicitly capturing the predictive signal provided by ApoE alleles.

## Discussion

These results demonstrate the potential of the Neomer approach to facilitate the development of sensitive and specific blood-based tests. By overcoming key limitations of traditional aptamer selection methods, this technology allows the same aptamer library to be applied in an agnostic manner to identify Aptamarkers for various disease states or medical conditions.

The successful application of Neomer-derived Aptamarkers for predicting brain amyloid deposition, a key pathological feature of Alzheimer's disease, has significant implications. Current methods for definitively diagnosing Alzheimer's, such as PET scans and CSF biomarker analysis, are invasive, expensive, and often impractical for widespread screening [24]. A blood test based on the Aptamarkers described here could provide a non-invasive, cost-effective, and widely accessible tool for early detection and monitoring of Alzheimer's disease. Early diagnosis is critical for implementing interventions that may slow disease progression [25].

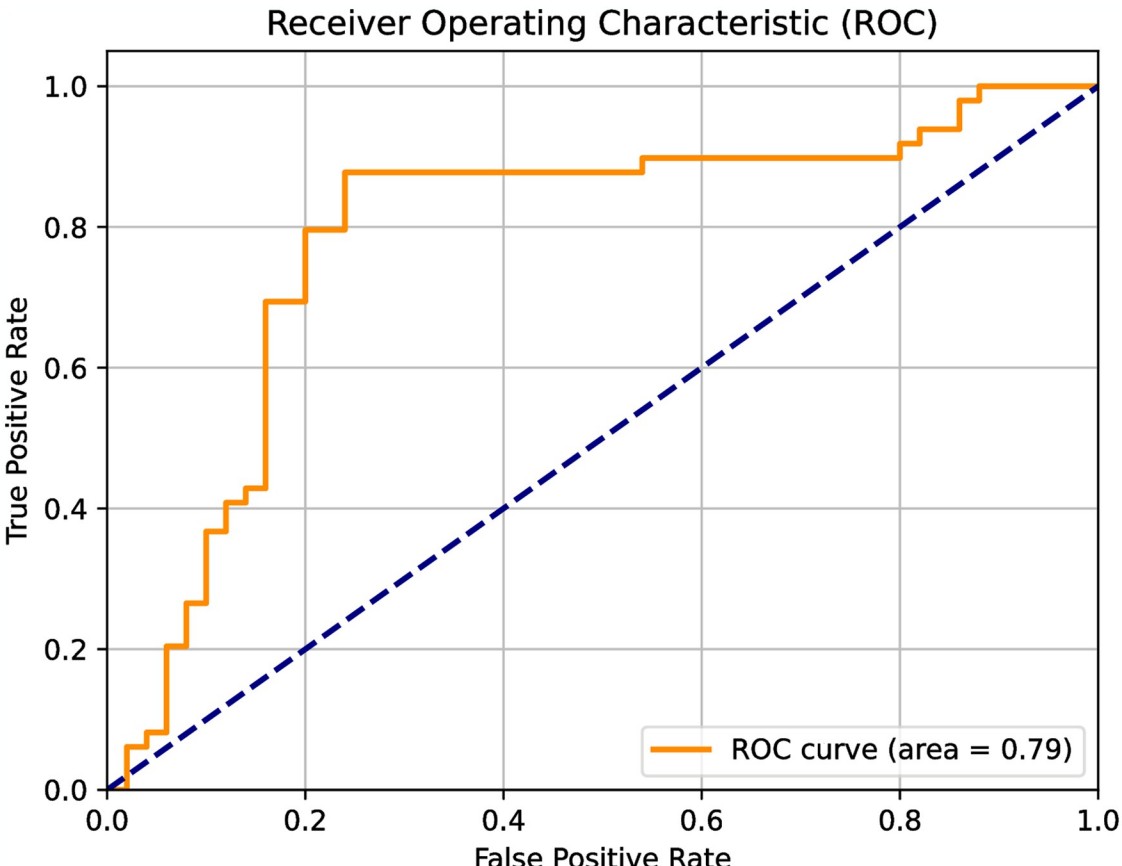

**Fig 6. Receiver operator response in the ET-1 predictive model.**

Importantly, we will soon be publishing results from a double-blind clinical trial further validating the performance of this Aptamarker-based blood test for Alzheimer's. This will provide additional evidence for the potential clinical implementation of this diagnostic technology.

Furthermore, by applying the Neomer approach to other diseases, it may be possible to develop novel blood-based diagnostic tests. The agnostic nature of the technology allows it to be adapted to different disease states without prior knowledge of specific biomarkers.

The Aptamarker approach represents an advance with the potential to improve disease diagnosis across a range of conditions. The successful development of a predictive blood test for Alzheimer's disease provides an example of the potential impact. With further research and validation, this platform technology could enable accessible, non-invasive, and accurate diagnostic testing, leading to earlier detection, timely intervention, and improved patient outcomes for numerous diseases. Ongoing studies will focus on clinical validation, translation, and expanding the application of this approach across diverse disease areas with unmet diagnostic needs.

## Supporting information

**S1 Fig. Distribution of clinical variables across the 390 individuals.**
(PDF)

**S1 Table. Clinical information of the 10 high and low amyloid individuals used for apta-marker selection.**
(CSV)

**S2 Table. Statistics and metrics of the models trained without ApoE.**
(CSV)

**S3 Table. Statistics and metrics of the models trained with ApoE.**
(CSV)

## Acknowledgments

We gratefully acknowledge the financial support from the Alzheimer's Drug Discovery Foundation (ADDF). We also thank the Australian Imaging, Biomarkers and Lifestyle (AIBL) study for providing valuable samples and guidance. The authors would like to thank Chloé Mansour and Kiera Drew of Neoventures Biotechnology Inc. for reviewing, editing, and formatting the manuscript.

## Author Contributions

**Conceptualization:** Gregory Penner.

**Data curation:** Cathal Meehan, Soizic Lecocq.

**Formal analysis:** Cathal Meehan, Gregory Penner.

**Methodology:** Soizic Lecocq, Gregory Penner.

**Software:** Cathal Meehan.

**Supervision:** Gregory Penner.

**Visualization:** Cathal Meehan.

**Writing – original draft:** Cathal Meehan, Soizic Lecocq, Gregory Penner.

**Writing – review & editing:** Cathal Meehan, Soizic Lecocq, Gregory Penner.

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
