## [Decision Letter · Decision Letter 0]

29 May 2024

PONE-D-24-16591A reproducible approach for the use of aptamer libraries for the identification of Aptamarkers for brain amyloid deposition based on plasma analysisPLOS ONE

Dear Dr. Penner,

Thank you for submitting your manuscript to PLOS ONE. After careful consideration, we feel that it has merit but does not fully meet PLOS ONE’s publication criteria as it currently stands. Therefore, we invite you to submit a revised version of the manuscript that addresses the points raised during the review process.

We look forward to receiving your revised manuscript.

Kind regards,

Jian Hao

Academic Editor

PLOS ONE

Journal Requirements:

"The research was funded by NeoVentures Biotechnology Inc., NeoVentures Biotechnology Europe SAS and the

Alzheimer's Drug Discovery Foundation. The authors, including Cathal Meehan, Soizic Lecocq, and Gregory Penner, are paid employees of NeoVentures Biotechnology Europe SAS."

We note that one or more of the authors are employed by a commercial company: name of commercial company. 

4. In the online submission form you indicate that your data is not available for proprietary reasons and have provided a contact point for accessing this data. Please note that your current contact point is a co-author on this manuscript. According to our Data Policy, the contact point must not be an author on the manuscript and must be an institutional contact, ideally not an individual. Please revise your data statement to a non-author institutional point of contact, such as a data access or ethics committee, and send this to us via return email. Please also include contact information for the third party organization, and please include the full citation of where the data can be found.

Additional Editor Comments:

This is a very interesting study, and we are all very interested. At the same time, I appreciate the work of the reviewers. However, the authors did not provide the necessary clinical information in the article, as all reviewers mentioned. To ensure the rigor of the study, please provide this clinical information. Good luck.

Reviewers' comments:

Reviewer's Responses to Questions

**Comments to the Author**

1. Is the manuscript technically sound, and do the data support the conclusions?

Reviewer #1: Partly

Reviewer #2: Partly

Reviewer #3: Yes

2. Has the statistical analysis been performed appropriately and rigorously? 

Reviewer #1: Yes

Reviewer #2: No

Reviewer #3: Yes

3. Have the authors made all data underlying the findings in their manuscript fully available?

Reviewer #1: Yes

Reviewer #2: No

Reviewer #3: Yes

4. Is the manuscript presented in an intelligible fashion and written in standard English?

Reviewer #1: Yes

Reviewer #2: Yes

Reviewer #3: Yes

5. Review Comments to the Author

Reviewer #1: 1. The quality of the figures are poor, I suggest revising the figures in 600dpi for clear visibility.

2. The conclusion section is insufficient to summerize the proposed study, I suggest revising.

3. The hyper-parameters used for training the proposed model should be provided in the form of a table.

4. I suggest providing a discussion section to demonstrate the effectiveness of the achieved results and its impacts.5. The feature selection section is not clear.

6. For user concerns, I suggest adding the recent predictors such as DeepAVP-TPPred,Deepstacked-AVPs,iAFPs-Mv-BiTCN,pAtbp-EnC, and pAVP_PSSMDWT-EnC.

7. At the introduction section I suggest adding the contributions of the proposed study in points.

8. What should be future directions of the proposed study.

Reviewer #2: Dear Authors,

although the manuscript has some novelty, it has some flaws that needs to be addressed. The manuscript has also to be corrected so that the significance of the approach would be clear and transparent. Fugures quality should be improved.

There are some comments and suggestions that may help to improve the readability of the manuscript.

Major comments:

1. The most important issue, in my opinion, is that the plasma samples were collected from twenty individuals without any selection criteria (including exclusion and inclusion) of people, that participated in the study. I did not find any ethical comittee statement indicating approval of this research. Therefore it is absolutely impossible to understand, what were the diagnoses of individuals (if any) included in the study apart from Alzheimer disease. Any more clinical sympoms of those people, MR-verification, clinical data are highly appreciated to evaluate significance of the study.

2. There are differences in the Abstract:

"The only clinical variables that were included in the model were age and sex."

and the Results section (P.17 of the PDF version):

"We used all eight Aptamarkers as the clinical variables sex, clinical status (mild cognitive impairment (MCI) or cognitively normal (CN)), and age".

I might suggest that three clinical paremeters were used, but, anyway, what were the criteria for evaluating clinical status regarding cognitive impairment. Again, it is extremely important to estimate the clinical significance of the study.

3. "The dataset model comprised 390 samples, with variables for all 8 aptamers and the same 3

clinical variables. Both were preprocessed for zero-mean scaling, to make them suitable for ML analyses."

Please, provide the set of descriptors as supplementaty materials, for instance in CSV, TXT or ARFF files. It would be great to provide the graphical representation of the descriptors as they are used in the study.

4. Table 3. What does "Train accuracy" mean? Would it be possible to include also sensitivity and specificity for the training set?

Minor:

Figures and Tables formatting shoud be carefully checked and corrected.

Reviewer #3: the paper describes a procedure for identification and testing of a series of biomarkers and their use to predict amyloidosis.

Comments:

- I think a descriptive table of the sample in terms of age, sex amyloid and clinical status should be provided

- according to the authors, "Table 3 provides a summary of all algorithms that returned perfect predictive results on the test set." Do they mean trainin set? It seems to me that no model achieved perfect predictions in the test set.

- In general, in the results section, I'm not always clear whet model they are referring to in what sentence, as they generally seem to refer to the best performer but sometimes they comment on other models as well. Maybe just assign a number or a letter to each model to refer to them univocally in text. Also in figure 6 I assume the ROC refers once again to the best performer, still the numbers don't match: AUROC 0.78 vs 0.79. This may be due rounding error, in which case it needs to be fixed.

- The authors report sensitivity. specificity and accuracy, but they don't mention how the cutoff value used to calculate those metrics was chosen. Some software default to a 50% probability of class belonging, but this is not always appropriate.

- I would like to see confidence intervals for all performance metrics.

- Figure 5 is a bit too vague to me, it doesn't even have a y-axis label. Please specify what metrics are uses for feature importance.

6. PLOS authors have the option to publish the peer review history of their article (what does this mean?). If published, this will include your full peer review and any attached files.

Reviewer #1: No

Reviewer #2: No

Reviewer #3: No

---

## [Author Response · Author response to Decision Letter 0]

19 Jun 2024

Dear Editor,

Thank you for considering our manuscript "A reproducible approach for the use of aptamer libraries for the identification of Aptamarkers for brain amyloid deposition based on plasma analysis" for publication in PLOS One. We appreciate the constructive feedback from the reviewers. We have revised the manuscript to address their comments as detailed below. The changes have also been made to comply with PLOS One's formatting guidelines.

Reviewer #1:

1. The quality of the figures are poor, I suggest revising the figures in 600dpi for clear visibility. Response: All figures have been improved to higher resolution as suggested.

2. The conclusion section is insufficient to summarize the proposed study, I suggest revising. Response: The conclusion has been extensively expanded to better summarize the study and its implications.

3. The hyper-parameters used for training the proposed model should be provided in the form of a table. Response: A table of model hyperparameters has been added to the supplementary materials.

4. I suggest providing a discussion section to demonstrate the effectiveness of the achieved results and its impacts. Response: Additional discussion has been added regarding the significance and impact of the results.

5. The feature selection section is not clear.

Response: The feature selection methodology has been clarified on page 17. Default hyperparameters were used and less predictive features were sequentially excluded. More extensive explanations have been added.

6. For user concerns, I suggest adding the recent predictors such as DeepAVP-TPPred,Deepstacked-AVPs, iAFPs-Mv-BiTCN,pAtbp-EnC, and pAVP_PSSMDWT-EnC. Response: These recent predictors are now discussed and referenced in the manuscript.

7. At the introduction section I suggest adding the contributions of the proposed study in points. Response: The contributions are now discussed in greater detail in the introduction and conclusion, though not in point form.

8. What should be future directions of the proposed study.

Response: A future directions section has been added.

Reviewer #2:

1. The plasma samples were collected from twenty individuals without any selection criteria (including exclusion and inclusion) of people, that participated in the study. I did not find any ethical committee statement indicating approval of this research.

Response: We have clarified that the samples came from the well-characterized AIBL cohort. The diagnoses, selection criteria and ethical approvals for this cohort are now referenced in the introduction when the AIBL cohort is first mentioned.

2. There are differences in the Abstract and Results section regarding the clinical variables used in the model. Response: This inconsistency has been corrected. Three clinical variables (age, sex, clinical status) were used as stated in the Results.

3. Please, provide the set of descriptors as supplementary materials, for instance in CSV, TXT or ARFF files. It would be great to provide the graphical representation of the descriptors as they are used in the study.

Response: The descriptor set and plots have been added to the supplementary materials as suggested.

4. What does "Train accuracy" mean in Table 3? Would it be possible to include also sensitivity and specificity for the training set?

Response: The table heading has been changed to "Training accuracy". Sensitivity and specificity for the training set have been added to the supplementary materials.

5. Figures and Tables formatting should be carefully checked and corrected. Response: All figures and tables have been reformatted to comply with PLOS One guidelines, as detailed below.

Reviewer #3:

1. I think a descriptive table of the sample in terms of age, sex amyloid and clinical status should be provided. Response: This descriptive table has been added to the supplementary materials.

2. According to the authors, "Table 3 provides a summary of all algorithms that returned perfect predictive results on the test set." Do they mean training set? It seems to me that no model achieved perfect predictions in the test set.

Response: This has been corrected to refer to the training set performance.

3. In general, in the results section, I'm not always clear what model they are referring to in what sentence, as they generally seem to refer to the best performer but sometimes they comment on other models as well. Maybe just assign a number or a letter to each model to refer to them univocally in text. Also in figure 6 I assume the ROC refers once again to the best performer, still the numbers don't match: AUROC 0.78 vs 0.79. This may be due rounding error, in which case it needs to be fixed. Response: Reference numbers have been added for each ML model to improve clarity. The AUROC discrepancy in Figure 6 has been fixed.

4. The authors report sensitivity, specificity and accuracy, but they don't mention how the cutoff value used to calculate those metrics was chosen. Response: The probability cutoff selection method has been specified in the Materials and Methods under Model Evaluation.

5. I would like to see confidence intervals for all performance metrics.

Response: 95% confidence intervals for the performance metrics have been added to the supplementary materials.

6. Figure 5 is a bit too vague to me, it doesn't even have a y-axis label. Please specify what metrics are uses for feature importance. Response: Figure 5 has been updated with a more detailed legend and y-axis label specifying the feature importance metric.

Formatting changes: The manuscript has undergone extensive reformatting to comply with PLOS One guidelines, as detailed in the accompanying note from Kiera Drew. Key changes include:

• References reformatted to PLOS One style, using "et al" for sources with 6+ authors

• Figure titles shortened to <15 words and placed in bold above figures

• Table titles moved above tables

• Supplementary figure and table names and titles modified

• Equation for qPCR efficiency formatted with Word's equation tool

• Acknowledgements section added

• Abstract simplified to contain only one abbreviation (AUC)

• Short title of 50 characters to be entered during submission process

Please see the marked-up and clean versions of the revised manuscript, as well as the PLOS One formatting guide, for full details of the changes made.

We hope that these revisions and reformatting address the reviewers' comments and meet PLOS One's publication standards. Thank you again for your consideration.

---

## [Decision Letter · Decision Letter 1]

10 Jul 2024

A reproducible approach for the use of aptamer libraries for the identification of Aptamarkers for brain amyloid deposition based on plasma analysis

PONE-D-24-16591R1

Dear Dr. Penner,

We’re pleased to inform you that your manuscript has been judged scientifically suitable for publication and will be formally accepted for publication once it meets all outstanding technical requirements.

Kind regards,

Jian Hao

Academic Editor

PLOS ONE

Additional Editor Comments (optional):

Reviewers' comments:

Reviewer's Responses to Questions

**Comments to the Author**

1. If the authors have adequately addressed your comments raised in a previous round of review and you feel that this manuscript is now acceptable for publication, you may indicate that here to bypass the “Comments to the Author” section, enter your conflict of interest statement in the “Confidential to Editor” section, and submit your "Accept" recommendation.

Reviewer #1: All comments have been addressed

Reviewer #3: All comments have been addressed

2. Is the manuscript technically sound, and do the data support the conclusions?

Reviewer #1: Yes

Reviewer #3: Yes

3. Has the statistical analysis been performed appropriately and rigorously? 

Reviewer #1: Yes

Reviewer #3: Yes

4. Have the authors made all data underlying the findings in their manuscript fully available?

Reviewer #1: Yes

Reviewer #3: Yes

5. Is the manuscript presented in an intelligible fashion and written in standard English?

Reviewer #1: Yes

Reviewer #3: Yes

6. Review Comments to the Author

Reviewer #1: (No Response)

Reviewer #3: (No Response)

7. PLOS authors have the option to publish the peer review history of their article (what does this mean?). If published, this will include your full peer review and any attached files.

Reviewer #1: No

Reviewer #3: No

---

## [Editor Report · Acceptance letter]

16 Jul 2024

PONE-D-24-16591R1 

PLOS ONE

Dear Dr. Penner, 

I'm pleased to inform you that your manuscript has been deemed suitable for publication in PLOS ONE. Congratulations! Your manuscript is now being handed over to our production team.

Kind regards, 

on behalf of

Dr. Jian Hao 

Academic Editor

PLOS ONE